# Characterizing gait pattern dynamics during symmetric and asymmetric walking using autoregressive modeling

Helia Mahzoun Alzakerin, Yannis Halkiadakis, Kristin D. Morgan *

Biomedical Engineering, School of Engineering, University of Connecticut, Storrs, Connecticut, United States of America

* Kristin.2.morgan@uconn.edu

**Data Availability Statement:** All relevant data are within the paper and its Supporting information files.

## Abstract

Gait asymmetry is often observed in populations with varying degrees of neuromuscular control. While changes in vertical ground reaction force (vGRF) peak magnitude are associated with altered limb loading that can be observed during asymmetric gait, the challenge is identifying techniques with the sensitivity to detect these altered movement patterns. Autoregressive (AR) modeling has successfully delineated between healthy and pathological gait during running; but has been little explored in walking. Thus, AR modeling was implemented to assess differences in vGRF pattern dynamics during symmetric and asymmetric walking. We hypothesized that the AR model coefficients would better detect differences amongst the symmetric and asymmetric walking conditions than the vGRF peak magnitude mean. Seventeen healthy individuals performed a protocol that involved walking on a split-belt instrumented treadmill at different symmetric (0.75m/s, 1.0 m/s, 1.5 m/s) and asymmetric (Side 1: 0.75m/s-Side 2:1.0 m/s; Side 1:1.0m/s-Side 2:1.5 m/s) gait conditions. Vertical ground reaction force peaks extracted during the weight-acceptance and propulsive phase of each step were used to construct a vGRF peak time series. Then, a second order AR model was fit to the vGRF peak waveform data to determine the AR model coefficients. The resulting AR coefficients were plotted on a stationarity triangle and their distance from the triangle centroid was computed. Significant differences in vGRF patterns were detected amongst the symmetric and asymmetric conditions using the AR modeling coefficients ($p = 0.01$); however, no differences were found when comparing vGRF peak magnitude means. These findings suggest that AR modeling has the sensitivity to identify differences in gait asymmetry that could aid in monitoring rehabilitation progression.

## Introduction

Gait asymmetry is often observed in individuals as a compensatory response to neuromuscular deficits [1–5]. Yet despite the fact that gait asymmetry is commonly exhibited in populations with varying degrees of neuromuscular control, differences in asymmetric gait patterns exist even amongst these affected groups [1–5]. Thus, it suggests that not all gait asymmetry patterns

**Funding:** The authors received no specific funding for this work.

**Competing interests:** The authors have declared that no competing interests exist.

are the same. However, uncovering these changes in gait patterns requires the identification of appropriate metrics and analysis techniques that possess the sensitivity to delineate between healthy and abnormal movement patterns. The vertical ground reaction force (vGRF) waveform is a valuable source to extract gait metrics from because it functions as a surrogate for one's center of mass motion with the vGRF peaks reflecting limb loading [4, 6, 7]. Previous studies observed that elevated vGRF peak magnitude and variability, particularly during the weight-acceptance and propulsion phases of gait, are often reported in individuals with detrimental joint loading and gait asymmetry [6–9]. Thus, this suggests that the investigation of vGRF peak gait patterns could yield valuable information about differences in gait dynamics.

Autoregressive (AR) modeling is a statistical technique that can evaluate waveform pattern variability to both quantitatively and visually denote changes in pattern dynamics [10, 11]. While a multitude of techniques can quantify gait pattern variability, AR modeling provides a way to graphically cluster times series with similar pattern dynamics together to help aid in the delineation between normal and abnormal movement patterns. The advantage of this approach is that the graphical component of this technique will allow researchers and clinicians to rapidly identify individuals with alternate and/or adverse gait patterns based on where they reside in the triangle. Often approaches compare the means between groups to assess gait differences; however, the problem is that comparing group means requires conducting studies that require large populations each time. Yet with AR modeling, while traditional studies with large populations will initially be needed to determine the regions on the AR triangle associated with specific conditions, once the regions have been defined researchers and clinicians can assess individuals gait based on their location in the triangle without the need to conduct a large scale study. Therefore, the AR modeling approach has the potential to serve as a valuable gait diagnostic tool. Thus, this study employed AR modeling to denote differences in symmetric and asymmetric walking pattern dynamics through both quantitative and graphical means as a first step to evaluate AR modeling's capability as a gait diagnostic tool.

Altered limb loading is often a characteristic of asymmetric gait and can be observed from the vGRF waveform peaks as they represent maximum loading on the limb [1, 3, 4, 7]. Furthermore, changes in vGRF peaks during walking have been associated with collagen degradation and synthesis in the knee; therefore, vGRF peak dynamics could potentially function as an important non-invasive measure of movement performance and joint health [12]. AR modeling is an ideal technique to quantify vGRF peak pattern dynamics because the AR model coefficients reveal the behavior and stationarity of the time series [10, 11]. Statistically, stationarity is used to determine if the mean, variance and autocorrelation structure of the waveform is constant over time; thus, in the context of gait analysis it can help evaluate the behavior of gait dynamics over time [10, 11]. Previous studies have successfully used AR modeling to assess differences in running dynamics in rearfoot runners [13], identify alterations in gait patterns in individuals with neurological deficits [14] and evaluate wheelchair propulsion asymmetry [15]. Morgan (2019) used second order AR modeling to identify differences in running dynamics between controls and post anterior cruciate ligament reconstruction (ACLR) individuals from vGRF peaks [16]. Their study showed that post ACLR individuals exhibited different between-limb running dynamics than controls based on their AR model coefficients which placed the groups in different regions of the stationarity triangle [16]. A substantial finding of this work was that significant differences in gait dynamics were detected using the AR model coefficients, yet no differences in vGRF peak magnitude were found between the two groups [16]. This suggests that AR modeling has the sensitivity to detect changes in pattern dynamics that may be difficult to quantify using traditional discrete metrics.

The objective of this study was to investigate how AR modeling could be used to both quantitatively and visually identify differences in gait pattern dynamics during symmetric and

asymmetric walking based on the graphical interpretation of the AR model analysis. Individuals walked at symmetric and two different asymmetric walking conditions with between-limb speed differences of 0.25m/s and 0.50 m/s to evaluate differences in gait dynamics from time series constructed from vGRF peaks. We hypothesized that the AR model coefficients will detect differences in gait patterns based on the vGRF peak patterns amongst the symmetric and asymmetric walking conditions better than an analysis of the vGRF peak magnitude means. We compared the AR model coefficients to the vGRF peak magnitude means because the vGRF peak means are often used to assess limb loading during gait and provide a standard metric to help understand, interpret and evaluate differences in gait dynamics. Overall employing AR modeling to differentiate between symmetric and asymmetric gait patterns could lead to this method being used more readily in rehabilitation to track the progression of changing gait dynamics.

## Materials and methods

### Instrumented gait analysis

Seventeen healthy participants (mean ± standard deviation; age: 20.8 ± 1.1 yrs; height: 1.68 ± 0.11 m; mass: 71.6 ± 10.9 kg; males:females: 10:7) performed a walking protocol. The participants ranged in age from 18 to 30 years old and provided written informed consent as required by the University of Connecticut institutional review board. All participants could not have had any previous knee surgery and were injury free for the past six months. Participants were asked to perform a walking protocol where walking speed and gait asymmetry were varied. First, participants performed a five-minute warm-up where they walked on the instrumented split-belt treadmill at a self-selected speed (Bertec Corporation, Columbus, Ohio). The participants' self-selected speed was determined by starting the participants at 1.0 m/s and then increasing the speed 0.1 m/s until the participant provided verbal feedback that the speed was their preferred self-selected speed. Once acclimated, participants walked at 0.75 m/s, 1.0 m/s, and 1.5 m/s, respectively for one minute each. These were the symmetric walking conditions because both limbs moved at the same speed. Then the participants walked with a between-limb gait speed difference of 0.25 m/s where the right limb was held at 1.0 m/s and the left limb was set to 0.75 m/s. The participants walked at this speed for one minute and then they once again walked with both limbs at 0.75 m/s for one minute to de-adapt. The asymmetric gait was repeated but this time the right limb was held at 1.0 m/s and the left limb was set to 1.5 m/s. The participants walked at this speed for one minute and then once again de-adapted at 0.75 m/s for one minute. The pattern of asymmetric gait followed by de-adaptation period was repeated on the left limb where it was held constant and the other was set to 0.75 m/s and 1.5 m/s, respectively. The gait speed was increased and decreased between trials at 0.1 m/s and then once at the desired speed individuals walked at least 10 strides before data was collected to reduce the potential of collecting spurious data. While the aforementioned gait protocol described how the asymmetric walking conditions were interspersed with de-adaptation, symmetric walking; we did randomize the order of the asymmetric walking trials. This protocol was adopted from previous studies [5, 17–19].

### Vertical ground reaction force peak extraction

Vertical GRF data were analyzed for each minute of the symmetric and asymmetric walking conditions. GRF data was collected from the instrumented treadmill at 1200 Hz and filtered at 35 Hz using a fourth-order Butterworth low-pass filter. Heel strike was denoted when 50N of force was detected from the vertical GRF. The vertical GRF waveform was normalized to body

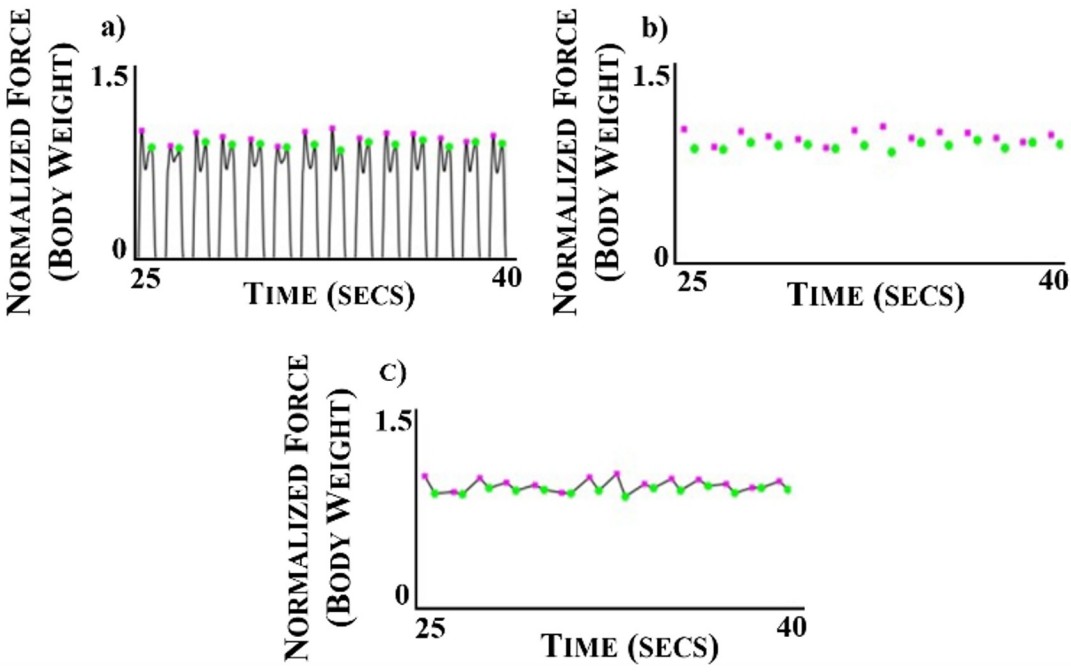

**Fig 1.** (a) Normalized vertical ground reaction force (GRF) waveform data collected from one individual while walking at 1.0 m/s during a 15 second time interval. The GRF peaks are denoted by the pink and green markers. (b) Normalized vertical GRF peaks extracted from the vertical GRF waveform. (c) Time series generated from the extracted vertical GRF peaks.

weight and the impact and second peaks were extracted from each step to construct the GRF peak patterns (Figs 1 and 2). The GRF peak extraction, the generation of the AR model and the stationarity analysis were performed using a custom MATLAB algorithm (MATLAB R2017a, The MathWorks, Inc., Natick, Massachusetts, USA).

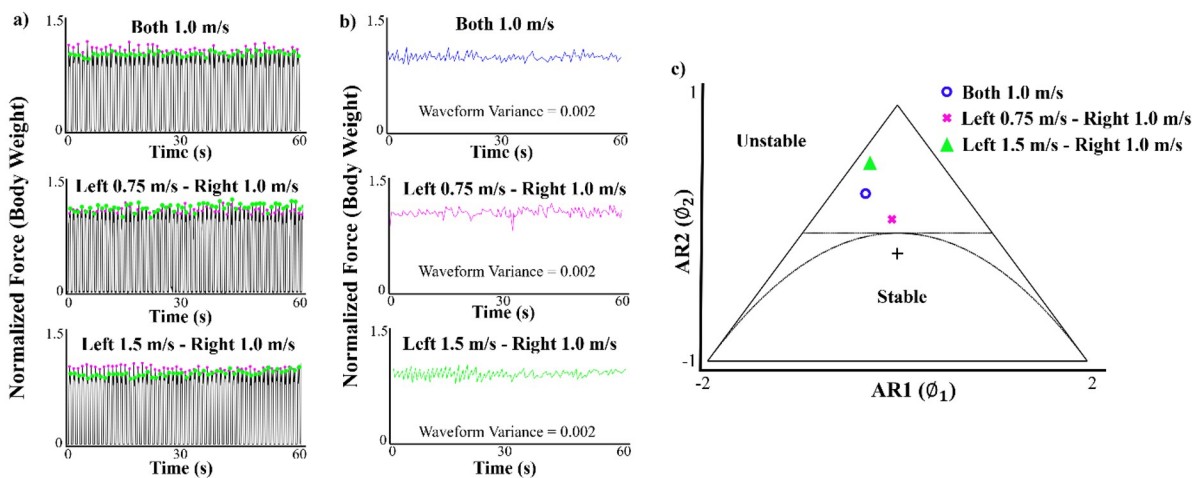

**Fig 2.** (a) Normalized vertical ground reaction force (GRF) waveforms for three different walking conditions during a one minute interval. The GRF peaks are denoted by the pink and green markers. (b) Plots of the vertical GRF peaks extracted and transformed into time series waveforms that were fit with second order autoregressive (AR) models. (c) Stationarity triangle with the locations of the second order AR model coefficients for the symmetric and asymmetric walking trials.

## Autoregressive modeling and analysis

Time series were generated for each participant from the normalized vertical GRF peaks extracted from the normalized vertical GRF waveform. Second order AR, AR(2), models were fit to the vertical GRF peaks time series' to evaluate the stationarity of the GRF peak pattern. The AR(2) models were fit to the data using an established procedure outlined by Box and Jenkins (1976) [10]. First, the mean was subtracted from the vertical GRF peak time series, which centers the data about zero and ensures that the AR model is fit to the data pattern. Then the autocorrelation (ACF) and partial autocorrelation (PACF) functions were used to determine the order of the AR model that best fit the vertical GRF peaks time series. The ACF and PACF plots indicate how well a time series is correlated with a delayed or lagged version of itself [10, 11]. The ACF and PACF plots supported that the AR(2) model was the best fit given that the ACF exhibited a slow decay while the PACF displayed a drop off after lag order 2. To further validate that an AR(2) was appropriate, the estimated model was plotted against the original time series (Fig 3a). The goodness of the AR(2) model fit was also evaluated by plotting the histogram of the residuals, which result from subtracting the estimated model time series from

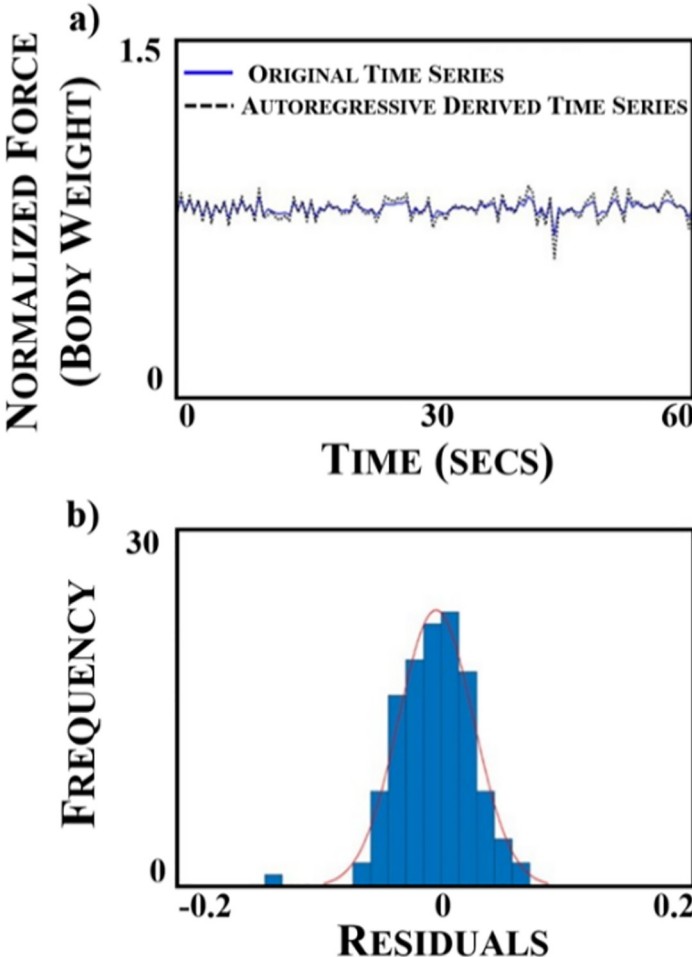

**Fig 3.** (a) Comparison of original normalized ground reaction force (GRF) time series plotted against a second order autoregressive, AR(2), derived times series fit to the original time series. (b) Histogram of the residuals obtained by subtracting the original and fitted waveforms.

the original time series. The histogram plots of the residuals were normally distributed about zero confirming that the AR(2) model was the appropriate model for the vertical GRF peak time series' [10, 11] (Fig 3b). The Anderson-Darling normality test was also conducted and indicated that the residuals were normally distributed. This model selection and validation criteria were applied to each GRF time series. Computing an R-squared value is not appropriate here because the time series model errors are not independent but serially correlated, therefore, we evaluated the residuals [10, 11].

An AR model models time series as a function of the previous values of the same time series. Specifically, an AR(2) model indicates that the times series is modeled as a function of the two previous values that proceed the current value (1).

$$y_t = \delta + \varnothing_1 y_{t-1} + \varnothing_2 y_{t-2} + \varepsilon_t \qquad (1)$$

Here $\varnothing_1$ and $\varnothing_2$ represent the AR model coefficients, $y_t$ is the current time series value, $y_{t-1}$ and $y_{t-2}$ represent the time series values at the two proceeding time intervals t-1 and t-2, $\delta$ is a constant and $\varepsilon_t$ is white noise. For an AR(2) model the stability for the time series can be determined from the model coefficients using the characteristic eq (2) and solving for the roots of the polynomial, $m_1$ and $m_2$ (3).

$$m^2 - \varnothing_1 m - \varnothing_2 = 0 \qquad (2)$$

$$m_1, m_2 = \frac{\varnothing_1 \pm \sqrt{\varnothing_1^2 + 4\varnothing_2}}{2} \qquad (3)$$

In lieu of having to perform these calculations the stationarity triangle provides a graphical and visual means to evaluate the stationarity of the time series. To assess the time series stationarity, the AR(2) coefficients, AR1 ($\varnothing_1$) and AR2 ($\varnothing_2$), are plotted on the x and y axes respectively, and if the resulting location of the coefficients resides inside of the triangle the time series is stationary (Montgomery et al. 2015; Box and Jenkins 1976). However, if the resulting point lies outside of the triangle the time series is unstable [10, 11]. While all points that lie inside of the triangle represent stable time series', a model is deemed as less stable as the point moves closer towards the edge of the triangle region. To quantify differences in model stability using the stationarity triangle, we computed the distance of the point from the centroid of the triangle (0, -1/3).

## Statistical analysis

A one-way ANOVA and Tukey post-hoc analysis was also performed to compare the mean magnitude of the vertical GRF peaks and distance of the AR model points from the centroid of the triangle for the symmetric and asymmetric gait conditions. All statistical analyses were conducted in SPSS (Version 23, IBM, Amonk, NY, USA.).

## Results

A comparison of the vertical GRF peak time series when both limbs were moving at 0.75m/s, 1.0 m/s and 1.5 m/s presented no differences in vertical GRF peak patterns (Fig 4a, Table 1). A comparison of the vertical GRF peak time series for when both limbs were moving at 1.0 m/s and the two asymmetric conditions determined that the individuals exhibited a significantly different vertical GRF peak pattern during the asymmetric walking condition than the symmetric walking condition (both limbs at 1.0 m/s) (p = 0.001) (Table 1). These differences were also observed on the stationarity triangle as the smaller between limb asymmetric walking conditions patterns were clustered towards the center, more stable region of the triangle (Fig 4b).

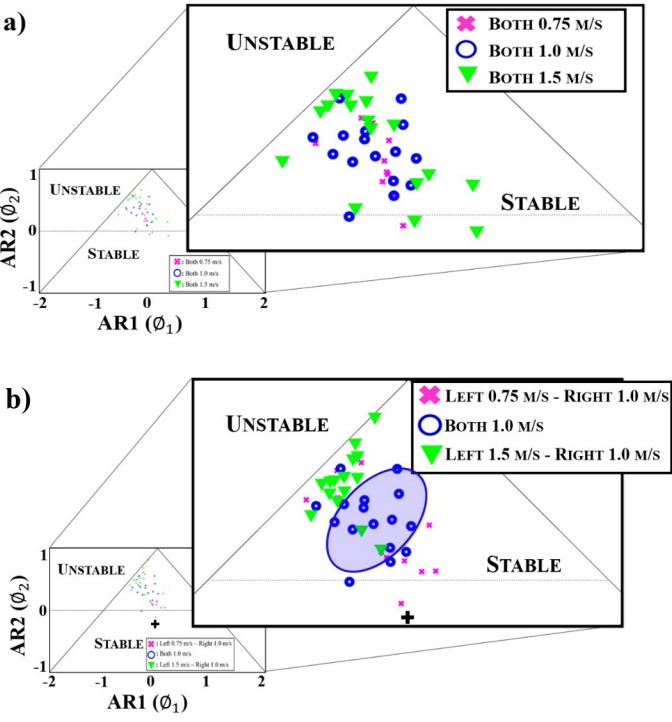

**Fig 4. Stationarity triangles where the AR1, $\varnothing_1$, and AR2, $\varnothing_2$, serve as the x and y axes, respectively.** (a) Zoomed in plot of the autoregressive (AR) model coefficients on the stationarity triangle for the three symmetric conditions when both limbs are moving at the same speed—0.75 m/s (pink x), 1.0 m/s (blue circle) and 1.5 m/s (green triangle). (b) Zoomed in plot of the AR model coefficients on the stationarity triangle for symmetric and asymmetric walking conditions. The blue oval encircles the region for the AR coefficients that represent the individuals walking with both limbs (symmetric) at 1.0 m/s. The + represents the centroid of the triangle from which the AR distances were computed.

**Table 1. Comparison of normalized vertical GRF (vGRF) peaks mean and distance at increasing walking speed and symmetric and asymmetric gait for each limb mean (standard deviation).**

| Comparison of Symmetric Gait | | | | |
|---|---|---|---|---|
| **Right Limb** | **Both 0.75 m/s** | **Both 1.0 m/s** | **Both 1.5 m/s** | **P-Value** |
| **Mean Peak vGRF** | 1.0 (0.0)[A] | 1.1 (0.1)[A] | 1.2 (0.1)[B] | <0.01 |
| **AR Distance** | 0.6 (0.2) | 0.7 (0.2) | 0.7 (0.2) | 0.23 |
| **Left Limb** | **Both 0.75 m/s** | **Both 1.0 m/s** | **Both 1.5 m/s** | **P-Value** |
| **Mean Peak vGRF** | 1.0 (0.0)[A] | 1.0 (0.1)[A] | 1.2 (0.1)[B] | <0.01 |
| **AR Distance** | 0.6 (0.3) | 0.6 (0.2) | 0.7 (0.2) | 0.26 |
| Comparison of Symmetric and Asymmetric Gait | | | | |
| **Right Limb** | **Both 1.0 m/s** | **L 0.75 m/s—R 1.0 m/s** | **L 1.5 m/s—R 1.0 m/s** | **P-Value** |
| **Mean Peak vGRF** | 1.1 (0.1) | 1.0 (0.0) | 1.1 (0.1) | 0.32 |
| **AR Distance** | 0.7 (0.2)[A] | 0.6 (0.3)[A] | 0.9 (0.2)[B] | <0.01 |
| **Left Limb** | **Both 1.0 m/s** | **L 1.0 m/s—R 0.75 m/s** | **L 1.0 m/s—R 1.5 m/s** | **P-Value** |
| **Mean Peak vGRF** | 1.0 (0.1) | 1.0 (0.0) | 1.0 (0.0) | 0.95 |
| **AR Distance** | 0.6 (0.2)[A,B] | 0.5 (0.2)[A] | 0.8 (0.1)[B] | 0.01 |

[A,B,C] represent means that do not share the same letter are significantly different ($\alpha$ = 0.05) across each row.

The p-values represent significant differences found when comparing values in the same row.

Conversely, the larger between limb asymmetric walking patterns were clustered at the edge of the triangle, closer to the unstable region (Fig 4b). The larger asymmetric walking condition patterns was located a significant distance away from the smaller asymmetric walking condition patterns ($p < 0.01$) (Fig 4b). The symmetric walking condition where both limbs were moving at 1.0 m/s was located in between the two asymmetric walking conditions (Fig 4b).

Differences in mean vertical GRF peak magnitude were also found when comparing the three symmetric walking speeds—0.75 m/s, 1.0 m/s, 1.5 m/s—under equivalent limbs speed conditions, but no differences were found when comparing the mean vertical GRF peak magnitudes across the asymmetric walking conditions. Thus, differences in mean vertical GRF peak magnitude is more variable during symmetric walking than asymmetric walking.

## Discussion

The objective of this study was to investigate how AR modeling both quantitatively and visually identified differences in gait pattern stability during symmetric and asymmetric walking. The results supported the hypothesis as the AR2 model coefficients were able to differentiate amongst the symmetric and two asymmetric walking conditions quantitatively and visually. The same level of differentiation was not achieved using the mean of the vGRF peaks. Moreover, the locations at which the different symmetric and asymmetric walking groups clustered provided additional insight about the stability of the gait patterns and individuals dynamic response to the asymmetric walking perturbations. The fact that the individuals resided in three different locations on the stationarity triangle suggests that these individuals adopted three different gait patterns in response to the symmetric and the two asymmetric walking conditions. While it may have been anticipated that the individuals would adopt three different vGRF peak patterns in response to the three different walking conditions, it is significant to note that the analysis of the vGRF peak magnitude means did not reveal any differences across the three walking conditions. This suggests that for this data, the AR modeling analysis displayed a greater sensitivity in delineating amongst the symmetric and asymmetric walking conditions than more traditional comparisons of mean peak vGRF data.

The advantage of the AR modeling approach is that it provided both quantitative and visual measures to identify differences in gait patterns during the symmetric and asymmetric walking conditions. Visually, these metrics were based on their location in the stationarity triangle. The findings from this study indicated that the participants adopted different gait patterns during the asymmetric walking than the symmetric walking. While often the asymmetric condition placed the individuals at the edge of the triangle near the unstable region, for the 0.25 m/s between-limb difference some individuals adopted gait patterns that were more stable that the symmetric condition. This is possibly attributed to the fact in response to this 0.25 m/s between-limb perturbation, the individuals restrict and constrain their motion resulting in a more stable gait pattern denoted by their placement near the center of the triangle. Given that individuals gait is inherently variable, a restricted, constrained gait pattern would reside in the more stable region on the AR triangle than normal symmetric walking. However, in the case of the larger asymmetric condition where the between-limb difference was 0.50 m/s, this perturbation appeared to be more destabilizing thus placing individuals at the edge of the triangle near the unstable region, which corresponded with larger distance metrics due to the fact that the individuals resided further away from the center of the triangle. Thus, the combination of the AR coefficients and stationarity triangle provided a visual way to both group and differentiate individuals by their gait patterns and we were able to quantify these differences based on their distance from the center of the triangle.

AR modeling has been successfully implemented to assess movement dynamics with those studies often using a first order AR model [13, 20]. In those studies, first order AR models were appropriate to capture differences in movement dynamics; however, a second order AR model was the optimal fit for this data. Differences in model selection could be attributed to the differences in the variables used to construct the time series data. Winter and Challis (2017) constructed their time series from vGRF peaks during running whereas this study used both the vGRF peaks during the weight-acceptance and propulsive phases of walking [13]. However, Kucynski (1999) used a second order AR model to assess postural stability from center of pressure displacement [21]. Here the second AR peak played a significant role in differentiating between the three walking conditions as there was vertical separation amongst the three groups with the asymmetric walking conditions flanking the symmetric condition. Significant differences between the asymmetric conditions were observed in the AR2 coefficient as smaller AR2 coefficients were measured for the asymmetric walking condition with the smaller between-limb speed difference. Interestingly, an increase in the magnitude of the AR2 coefficients corresponded with a shift towards the unstable region. The damped or diminished presence of the AR2 coefficient during the smaller between-limb speed difference asymmetric walking trials compared to the larger asymmetric between-limb speed difference could suggest that the emergence of strong AR2 coefficient contribution could represent a shift toward more unstable gait dynamics. While additional work is needed to substantiate this observation, the results of this study highlight the importance of model selection and support the use of second order AR model to delineate amongst symmetric and asymmetric gait patterns.

A limitation of this work was that gait was analyzed during one-minute intervals. Traditional temporal analyses tend to require large time series of five minutes or more to evaluate temporal patterns [20, 22]. These alternate techniques are appropriate in their use of extended time series; however, AR modeling does not require such extensive time series for its analysis [11]. While tracking gait changes over larger time intervals could further delineate differences in gait patterns, we were able to successfully identify three different motor control strategies from the one-minute intervals. The ability to extract such information in a short time period suggests that this analysis technique could be used as a diagnostic tool to identify altered gait strategies and neuromuscular function.

## Conclusion

Autoregressive modeling can be a valuable tool to aid in the differentiation of gait pattern dynamics and stability via both quantitative and visual means. The location of the AR model coefficients on the stationarity triangle provided insight about changes in gait pattern dynamics based on the vertical GRF peak patterns individuals adopted in response to the gait disturbances. The results indicated that individuals adopted different gait patterns during the symmetric and asymmetric walking conditions. Given this techniques ability to differentiate between altered gait patterns, it suggests that AR modeling could potentially serve as a valuable clinical tool to help monitor injury and rehabilitation progression.

## Supporting information

**S1 Data.**
(XLSX)

## Author Contributions

**Conceptualization:** Kristin D. Morgan.

**Formal analysis:** Kristin D. Morgan.

**Investigation:** Helia Mahzoun Alzakerin, Yannis Halkiadakis, Kristin D. Morgan.

**Methodology:** Helia Mahzoun Alzakerin, Yannis Halkiadakis, Kristin D. Morgan.

**Writing – original draft:** Helia Mahzoun Alzakerin, Yannis Halkiadakis, Kristin D. Morgan.

**Writing – review & editing:** Helia Mahzoun Alzakerin, Kristin D. Morgan.

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
