## [Decision Letter · Decision Letter 0]

10 Sep 2020

PONE-D-20-21892

Characterizing Gait Pattern Dynamics during Symmetric and Asymmetric Walking using Time Series Analysis

PLOS ONE

Dear Dr. Morgan,

Thank you for submitting your manuscript to PLOS ONE. After careful consideration, we feel that it has merit but does not fully meet PLOS ONE’s publication criteria as it currently stands. Therefore, we invite you to submit a revised version of the manuscript that addresses the points raised during the review process.

Overall the reviewers expressed some enthusiasm for this manuscript, but highlighted several areas that will benefit from additional clarity.  Please carefully address each comment in your revision.

We look forward to receiving your revised manuscript.

Kind regards,

Eric R. Anson

Academic Editor

PLOS ONE

Journal Requirements:

Reviewers' comments:

Reviewer's Responses to Questions

**Comments to the Author**

1. Is the manuscript technically sound, and do the data support the conclusions?

Reviewer #1: Yes

Reviewer #2: Yes

2. Has the statistical analysis been performed appropriately and rigorously? 

Reviewer #1: No

Reviewer #2: Yes

3. Have the authors made all data underlying the findings in their manuscript fully available?

Reviewer #1: Yes

Reviewer #2: Yes

4. Is the manuscript presented in an intelligible fashion and written in standard English?

Reviewer #1: Yes

Reviewer #2: Yes

5. Review Comments to the Author

Reviewer #1: In this manuscript, the authors use vertical ground reaction forces to differentiate symmetric and asymmetric walking patterns. They induce asymmetric walking using a split-belt treadmill. Their autoregressive (AR) modeling revealed differences between the symmetric and asymmetric walking conditions.

TITLE

I suggest that the title could be made clearer with revision, as “time series analysis” is quite vague (i.e., many different variables could be subject to time series analysis when investigating gait dynamics).

ABSTRACT

Lines 33 and 34: the authors suggest that new methods are needed to detect to detect changes in vertical GRFs during asymmetric gait, but also state that these changes are “commonly observed”. These statements seem to contradict one another.

Line 42: I suggest using a colon rather than a dash to indicate the split-belt treadmill speeds.

Line 44-45: this sentence requires revision for grammar and punctuation.

INTRODUCTION

The justification for this study is somewhat unclear to me. While I certainly agree with the authors that it is important to be able to quantify the asymmetry of gait in clinical populations. there are already many well-established ways to do this. It would be helpful if the authors could provide some perspective about why this particular approach may be better than existing approaches, or at least how this approach may provide unique information relative to others.

The authors provide a broad introduction with regard to gait asymmetry, but it seems as though the primary interest is in leg loading and likely patients with knee injuries (knee injury is listed as specific exclusion criteria). If this is true, I suggest that they could provide a more focused introduction about how this technique could specifically be useful for certain populations.

They authors make a couple of references to the ability of AR modeling to “visually” detect differences in gait parameters. Further clarification would be helpful, as it is not clear to me what this means or why it is useful to detect a result “visually” when the same result can be expressed quantitatively.

METHODS

Line 115 – how was self-selected speed determined?

Lines 159-161 – was any quantitative analysis performed to determine that AR(2) was the best fit model or was this done by visual inspection? A quantitative analysis would provide much stronger support for this model selection.

RESULTS

Were the left and right legs subjected to separate analyses? This is appears to be the case in Table 1. If the authors wish to analyze the legs separately, I suggest a two-way (leg x condition) ANOVA.

DISCUSSION

No comments.

Reviewer #2: 1. Abstract

a. Primary analysis: autoregressive modeling (previously used for running but not walking).

b. Hypothesize that autoregressive model coefficients better detect gait asymmetries than peak vGRF magnitude.

c. Used second order AR model

d. The asymmetric condition notation is confusing and could be presented more clearly. Perhaps (Side 1: XXX & Side 2: XXX).

2. Introduction

a. Gave good justification for the importance of quantifying gait asymmetry.

b. Gave compelling reasons for using AR and its success in previous studies.

c. Objective: “investigate how AR modeling could be used to both quantitatively and visually identify differences in gait pattern dynamics during symmetric and asymmetric walking.”

i. Can you please clarify if by “visually”, you are referring to graphical interpretations or clinical assessments?

ii. Induced asymmetric walking with instrumented split belt treadmill.

d. Hypothesis: ARM model coefficients will detect differences in giat patterns based on vGRF peak patterns than peak mean vGRF magnitudes.

i. Did you consider comparing RMS differences of interlimb vGRF?

ii. Are there other measures that may isolate differences in separate phases of the gait cycle? Although mean peak vGRF is often used, it’s easy to see that it may not be the most sensitive measure. It might be good to address why you chose to only consider the mean magnitude.

3. Methods

a. Gait Analysis

i. Were the trial orders randomized? If not, how do you mitigate potential learning effects?

ii. Line 118: “walked at an asymmetric gait” is awkward.

iii. From my understanding, individuals only have 10 strides in an asymmetric condition before data is collected. Is the goal to collect data in a transient or steady state condition? If steady-state is the goal, I would be hesitant to say this is steady-state.

b. Peak vGRF Extraction

c. Autoregressive Modeling & Analysis

i. Modeling was described very clearly!

d. Statistical Analysis

4. Results

a. Figure 4b: interesting that some people have “more stable” gait patterns in the 0.75 m/s Left condition than they are in the symmetric condition. Do you have an explanation for that?

b. Lines 233 -235: are these interlimb differences, differences across conditions, or differences between your calculation methods. Would be helpful to explain.

c. Table 1: Can you provide more explanation to make the information in this table more clear?

i. Are these Mean and distance parameters form the AR models? Or are they differences in mead vGRF magnitude?

ii. Which comparisons are the p values for? Is this symmetric vs both asymmetric?

iii. I see the note about ABC for grouping variables that are not significantly different from one another. It might be more useful to just see the asymmetric conditions compared back to the symmetric.

5. Discussion

a. Lines 244 – 249 & figure 4b: I see pretty clear clusters for the left 1.5 and symmetric conditions, but the left 0.75 seems to be all over the place. How do you explain that?

b. Discussion section would benefit from a more thorough comparison of the different symmetry conditions and model parameters. For example, discussing why some models were more or less stable. Did the AR modeling differentiate all asymmetric conditions from the symmetric? Was this method able to differentiate differences between the asymmetric conditions? Are there any clear metrics or measures that a clinical could take away from this when treating patients (i.e. if they are asymmetric, but “more stable” than most symmetric adults, how does that affect their treatments)?

6. Conclusions

a. I agree with the overall conclusion and think this work supports it. However, it would be nice to have more clarification/interpretation of the results between numbers on a graph and the big picture (AR modeling is good) at the end.

7. Figures

a. The figures appear blurry in the pdf but clear in the download. Be sure to confirm hat the clear ones are used in the final publication.

b. Table 1: It’s best practice to keep significant figure or decimal places consistent in tables.

6. PLOS authors have the option to publish the peer review history of their article (what does this mean?). If published, this will include your full peer review and any attached files.

Reviewer #1: No

Reviewer #2: No

---

## [Author Response · Author response to Decision Letter 0]

30 Oct 2020

Review: PONE-S-20-27379

Title: "Characterizing Gait Pattern Dynamics during Symmetric and Asymmetric Walking using Time Series Analysis"

We thank the Editor and Reviewers for the opportunity to revise the manuscript. The revised manuscript provides stronger justification of the metrics used for the analysis as well as greater clarity regarding the interpretation of the results. We believe these changes make for a clearer and more concise manuscript and we thank the Reviewers for taking the time to review the manuscript.

Reviewer #1: In this manuscript, the authors use vertical ground reaction forces to differentiate symmetric and asymmetric walking patterns. They induce asymmetric walking using a split-belt treadmill. Their autoregressive (AR) modeling revealed differences between the symmetric and asymmetric walking conditions.

TITLE

I suggest that the title could be made clearer with revision, as “time series analysis” is quite vague (i.e., many different variables could be subject to time series analysis when investigating gait dynamics).

Thank you for this suggestion. We have revised the title and substituted autoregressive modeling for time series analysis.

ABSTRACT

Lines 33 and 34: the authors suggest that new methods are needed to detect to detect changes in vertical GRFs during asymmetric gait, but also state that these changes are “commonly observed”. These statements seem to contradict one another.

Yes, thank you for identifying this. We have revised the sentence to address the contradictory nature of the sentence (Line 35).

Line 42: I suggest using a colon rather than a dash to indicate the split-belt treadmill speeds.

Thank you. The dashed lines have been removed (Line 42).

Line 44-45: this sentence requires revision for grammar and punctuation.

Agreed, the sentence was not clear, and we have modified it to provide greater clarity (Lines 44-45). 

INTRODUCTION

The justification for this study is somewhat unclear to me. While I certainly agree with the authors that it is important to be able to quantify the asymmetry of gait in clinical populations. there are already many well-established ways to do this. It would be helpful if the authors could provide some perspective about why this particular approach may be better than existing approaches, or at least how this approach may provide unique information relative to others.

The authors provide a broad introduction with regard to gait asymmetry, but it seems as though the primary interest is in leg loading and likely patients with knee injuries (knee injury is listed as specific exclusion criteria). If this is true, I suggest that they could provide a more focused introduction about how this technique could specifically be useful for certain populations.

They authors make a couple of references to the ability of AR modeling to “visually” detect differences in gait parameters. Further clarification would be helpful, as it is not clear to me what this means or why it is useful to detect a result “visually” when the same result can be expressed quantitatively.

Yes, the previous version of the manuscript did not provide a clear justification for the purpose and advantage of AR modeling. The introduction has been revised to address this oversight (Lines 75 – 86 and Line 108). 

METHODS

Line 115 – how was self-selected speed determined?

We have revised the manuscript to include greater detail about how we determined the participants self-selected speed (Lines 129-131).

Lines 159-161 – was any quantitative analysis performed to determine that AR(2) was the best fit model or was this done by visual inspection? A quantitative analysis would provide much stronger support for this model selection.

Yes, a quantitative analysis was performed to evaluate if the AR(2) model was the best model to fit to the data. The appropriateness of the model fit was discussed in lines 180-185 and Figure 3b. However, we also updated the manuscript to also include why computing the R-Squared is not appropriate for this analysis; however, the results of the Anderson-Darling normality test yielded a p-value, a quantitative metric, that allowed us to confirm that the residuals were normally distributed (Lines 185-187).

RESULTS

Were the left and right legs subjected to separate analyses? This is appears to be the case in Table 1. If the authors wish to analyze the legs separately, I suggest a two-way (leg x condition)ANOVA.

We ran a two-way ANOVA that compared the leg x condition and it showed that the results were not altered based on the limb. It was consistent the AR distance metric was different between the symmetric and asymmetric condition for both the right and left limb, meaning the results were the same for both limbs.

DISCUSSION

No comments.

Reviewer #2: 

1. Abstract

a. Primary analysis: autoregressive modeling (previously used for running but not walking).

b. Hypothesize that autoregressive model coefficients better detect gait asymmetries than peak vGRF magnitude.

c. Used second order AR model

d. The asymmetric condition notation is confusing and could be presented more clearly. Perhaps (Side 1: XXX & Side 2: XXX).

Yes, the asymmetric walking notation was confusing and has been corrected in the manuscript (Lines 42-43).

2. Introduction

a. Gave good justification for the importance of quantifying gait asymmetry.

b. Gave compelling reasons for using AR and its success in previous studies.

c. Objective: “investigate how AR modeling could be used to both quantitatively and visually identify differences in gait pattern dynamics during symmetric and asymmetric walking.”

i. Can you please clarify if by “visually”, you are referring to graphical interpretations or clinical assessments?

By visual, we were referring to the graphical interpretation of the results and revised the manuscript to clarify that (Line 108).

ii. Induced asymmetric walking with instrumented split belt treadmill.

d. Hypothesis: ARM model coefficients will detect differences in giat patterns based on vGRF peak patterns than peak mean vGRF magnitudes.

i. Did you consider comparing RMS differences of interlimb vGRF?

We did not consider comparing RMS differences of interlimb vGRF. We decided on comparing the AR modeling to the mean of the vGRF peak magnitude because that is readily used metric for analyzing gait. However, we acknowledge that RMS could have also been used and will consider using RMS for future studies.

ii. Are there other measures that may isolate differences in separate phases of the gait cycle? Although mean peak vGRF is often used, it’s easy to see that it may not be the most sensitive measure. It might be good to address why you chose to only consider the mean magnitude.

Yes, that is true. We should have discussed why we decided to use the mean peak vGRF and have included an explanation in the manuscript (Lines 113-115).

3. Methods

a. Gait Analysis

i. Were the trial orders randomized? If not, how do you mitigate potential learning effects?

Yes, the trial orders were randomized. We did not mention that in the previous version of the manuscript, but the revised version of the manuscript does acknowledge that the trials were randomized (Lines 142-144). 

ii. Line 118: “walked at an asymmetric gait” is awkward.

This sentence has been corrected to provide greater clarity (Line 133).

iii. From my understanding, individuals only have 10 strides in an asymmetric condition before data is collected. Is the goal to collect data in a transient or steady state condition? If steady-state is the goal, I would be hesitant to say this is steady-state.

Yes, we agree the individuals would not reach a steady-state condition after 10 strides and we did not want them to reach a steady-state. Our goal was to capture the dynamics of individuals during the various asymmetric and symmetric walking conditions. Capturing the data after 10 strides is a standard operating procedure in our laboratory, and in the case of this study eliminated any major stumbles that can occur due to changing the speed of the treadmill. But we agree that the participants would not have reached a steady-state after 10 strides and no it was not the goal that they reached steady state for this analysis.

b. Peak vGRF Extraction

c. Autoregressive Modeling & Analysis

i. Modeling was described very clearly!

d. Statistical Analysis

4. Results

a. Figure 4b: interesting that some people have “more stable” gait patterns in the 0.75 m/s Left condition than they are in the symmetric condition. Do you have an explanation for that?

Yes, we believe that the Left: 0.75 m/s – Right 1.0 m/s asymmetric walking condition was more stable than the symmetric walking condition because during the smaller 0.25 m/s asymmetry the participants adopted a more conservative, restrictive gait pattern. They likely adopt this conservative gait pattern because they recognize that they are experiencing an unstable perturbation they try to minimize the instability by restricting their motion. However, during normal walking individuals there is an inherent level of gait variability because individuals gait needs to be “pliable” and adapt to various, unexpected perturbations. Thus, in order to be able to adapt to different gait patterns they cannot walk in a restrictive motion which likely contributes to greater variability in their pattern than the 0.25 m/s between-limb gait speed difference condition. 

b. Lines 233 -235: are these interlimb differences, differences across conditions, or differences between your calculation methods. Would be helpful to explain.

Yes, that was not clear and we revised that section to better explain what the differences were comparing (Lines 255-259).

c. Table 1: Can you provide more explanation to make the information in this table more clear?

i. Are these Mean and distance parameters form the AR models? Or are they differences in mead vGRF magnitude?

The mean was for the vertical GRF peak magnitudes and the distance was for the AR model. This was not clear and Table 1 has been updated to address this and the other concerns raised.

ii. Which comparisons are the p values for? Is this symmetric vs both asymmetric?

The p-values are for the data in the specific row so for the first set, we are comparing the means for the vertical GRF peak magnitude across the three symmetric conditions – 0.75 m/s, 1.0 m/s, and 1.5 m/s for the right leg. Thus, the p-value at the end of the row represents the comparison of the data in that row. 

iii. I see the note about ABC for grouping variables that are not significantly different from one another. It might be more useful to just see the asymmetric conditions compared back to the symmetric.

I apologize if we are not understanding the question. Our intent was to show differences in the mean vertical GRF peak magnitudes across the different gait conditions as well as to show the differences in AR distance metric across the different gait conditions. 

5. Discussion

a. Lines 244 – 249 & figure 4b: I see pretty clear clusters for the left 1.5 and symmetric conditions, but the left 0.75 seems to be all over the place. How do you explain that?

This is a great question and we revised the manuscript and added a paragraph to the discussion to explain this (Lines 276-292).

b. Discussion section would benefit from a more thorough comparison of the different symmetry conditions and model parameters. For example, discussing why some models were more or less stable. Did the AR modeling differentiate all asymmetric conditions from the symmetric? Was this method able to differentiate differences between the asymmetric conditions? Are there any clear metrics or measures that a clinical could take away from this when treating patients (i.e. if they are asymmetric, but “more stable” than most symmetric adults, how does that affect their treatments)?

These are all great questions and we did our best to address them in the additional paragraphs added to the discussion.

6. Conclusions

a. I agree with the overall conclusion and think this work supports it. However, it would be nice to have more clarification/interpretation of the results between numbers on a graph and the big picture (AR modeling is good) at the end.

We worked to provide greater clarity between the quantitative and graphical measures in Lines 276-292 as well as Lines 325-326.

7. Figures

a. The figures appear blurry in the pdf but clear in the download. Be sure to confirm hat the clear ones are used in the final publication.

Thank you, we will work with the journal to make sure the best quality figures are used.

b. Table 1: It’s best practice to keep significant figure or decimal places consistent in tables.

Yes, you are correct, and Table 1 has been adjusted to correct this error.

---

## [Decision Letter · Decision Letter 1]

18 Nov 2020

Characterizing Gait Pattern Dynamics during Symmetric and Asymmetric Walking using Autoregressive Modeling

PONE-D-20-21892R1

Dear Dr. Morgan,

We’re pleased to inform you that your manuscript has been judged scientifically suitable for publication and will be formally accepted for publication once it meets all outstanding technical requirements.

Kind regards,

Eric R. Anson

Academic Editor

PLOS ONE

Additional Editor Comments (optional):

Reviewers' comments:

Reviewer's Responses to Questions

**Comments to the Author**

1. If the authors have adequately addressed your comments raised in a previous round of review and you feel that this manuscript is now acceptable for publication, you may indicate that here to bypass the “Comments to the Author” section, enter your conflict of interest statement in the “Confidential to Editor” section, and submit your "Accept" recommendation.

Reviewer #1: All comments have been addressed

2. Is the manuscript technically sound, and do the data support the conclusions?

Reviewer #1: Yes

3. Has the statistical analysis been performed appropriately and rigorously? 

Reviewer #1: Yes

4. Have the authors made all data underlying the findings in their manuscript fully available?

Reviewer #1: Yes

5. Is the manuscript presented in an intelligible fashion and written in standard English?

Reviewer #1: Yes

6. Review Comments to the Author

Reviewer #1: The authors have addressed my prior comments. I have no further suggestions and thank the authors for sharing their nice work.

7. PLOS authors have the option to publish the peer review history of their article (what does this mean?). If published, this will include your full peer review and any attached files.

Reviewer #1: No

---

## [Editor Report · Acceptance letter]

23 Nov 2020

PONE-D-20-21892R1 

Characterizing Gait Pattern Dynamics during Symmetric and Asymmetric Walking using Autoregressive Modeling 

Dear Dr. Morgan:

I'm pleased to inform you that your manuscript has been deemed suitable for publication in PLOS ONE. Congratulations! Your manuscript is now with our production department. 

Kind regards, 

on behalf of

Dr. Eric R. Anson 

Academic Editor

PLOS ONE